# Loop-Mediated Isothermal Amplification (LAMP) and SYBR Green qPCR for Fast and Reliable Detection of *Geosmithia morbida* (Kolařik) in Infected Walnut

**DOI:** 10.3390/plants11091239

**Published:** 2022-05-03

**Authors:** Domenico Rizzo, Chiara Aglietti, Alessandra Benigno, Matteo Bracalini, Daniele Da Lio, Linda Bartolini, Giovanni Cappellini, Antonio Aronadio, Cristina Francia, Nicola Luchi, Alberto Santini, Santa Olga Cacciola, Tiziana Panzavolta, Salvatore Moricca

**Affiliations:** 1Laboratory of Phytopathological Diagnostics and Molecular Biology, Plant Protection Service of Tuscany, Via Ciliegiole 99, 51100 Pistoia, Italy; domenico.rizzo@regione.toscana.it (D.R.); linda.bartolini@regione.toscana.it (L.B.); giovanni.cappellini@regione.toscana.it (G.C.); cristina.francia@regione.toscana.it (C.F.); 2Department of Agricultural, Food, Environmental and Forestry Science and Technology (DAGRI), Plant Pathology and Entomology Section, University of Florence, Piazzale delle Cascine 28, 50144 Florence, Italy; chiara.aglietti@unifi.it (C.A.); alessandra.benigno@unifi.it (A.B.); matteo.bracalini@unifi.it (M.B.); tiziana.panzavolta@unifi.it (T.P.); 3Department of Agricultural, Food and Agro-Environmental Sciences, University of Pisa, Via del Borghetto 80, 56124 Pisa, Italy; daniele.dalio@hotmail.com; 4Plant Protection Service of Tuscany, c/o Interporto Toscano “Amerigo Vespucci”, Collesalvetti, 57014 Livorno, Italy; antonio.aronadio@regione.toscana.it; 5Institute for Sustainable Plant Protection, National Research Council (IPSP-CNR), Via Madonna del Piano 10, 50019 Sesto Fiorentino, Italy; nicola.luchi@cnr.it (N.L.); alberto.santini@cnr.it (A.S.); 6Department of Agriculture, Food and Environment, University of Catania, 95123 Catania, Italy; olgacacciola@unict.it

**Keywords:** ascomycete fungus, xylophagous insect, quarantine organisms, molecular identification, diagnostic tools, phytosanitary monitoring, disease surveillance

## Abstract

Walnut species (*Juglans* spp.) are multipurpose trees, widely employed in plantation forestry for high-quality timber and nut production, as well as in urban greening as ornamental plants. These species are currently threatened by the thousand cankers disease (TCD) complex, an insect–fungus association which involves the ascomycete *Geosmithia morbida* (GM) and its vector, the bark beetle *Pityophthorus juglandis*. While TCD has been studied extensively where it originated in North America, little research has been carried out in Europe, where it was more recently introduced. A key step in research to cope with this new phytosanitary emergency is the development of effective molecular detection tools. In this work, we report two accurate molecular methods for the diagnosis of GM, based on LAMP (real-time and visual) and SYBR Green qPCR, which are complimentary to and integrated with similar recently developed assays. Our protocols detected GM DNA from pure mycelium and from infected woody tissue with high accuracy, sensitivity, and specificity, without cross-reactivity to a large panel of taxonomically related species. The precision and robustness of our tests guarantee high diagnostic standards and could be used to support field diagnostic end-users in TCD monitoring and surveillance campaigns.

## 1. Introduction

The family Juglandaceae DC. ex Perleb comprises several tree species distributed across North and South America and Eurasia, where they occur in forest ecosystems and productive plantations [1,2]. Within this family, members of the genus *Juglans* have economic and ecological importance as they contribute food and shelter to wildlife, as well as food and wood for humans. Some species, such as *Juglans nigra* L., commonly referred to as black walnut or eastern black walnut, and *Juglans regia* L., also called English walnut or Persian walnut, have become popular for their high-quality timber and edible nuts, and are widely cultivated in tree farming activities. Because of the nutraceutical properties of their highly prized fruit, commercial nut production sustains a thriving economy with an expanding market [3].

Black walnut, traditionally known for its resistance to pathogens and pests [4], has in recent years been strongly threatened by a major phytosanitary constraint: the thousand cankers disease of walnut (TCD) [5,6,7]. The disease is caused by the lethal association between the ascomycete *Geosmithia morbida* (GM) Kolařik (Hypocreales, Bionectriaceae) and the insect vector *Pityophthorus juglandis* (PJ) Blackman (Coleoptera, Curculionidae, Scolytinae) [5]. This emerging disease originated approximately two decades ago, when widespread decline and mortality of black walnut were observed in Boulder and Colorado Springs (CO, USA). The disease starts with a general decline in walnut trees, accompanied by yellowing and thinning of leaves in the higher portions of the crown. After some years, symptoms such as wilting and dieback of the crown with branch mortality, become noticeable. This pattern of disease progression depends on a gradual intensification of the attack by this insect–fungus complex, which extends from the upper crown basipetally to lower larger branches, and to the stem [8]. The native range of TCD was initially restricted to the southwestern states of the USA, where it aggressively attacked the exotic black walnut. The disease subsequently spread eastward to the black walnut native range, including many states in the eastern USA [2]. At present, the only European country in which TCD has been reported is Italy [9,10], which represents the current eastern distribution range of the disease. GM and PJ are regulated as quarantine organisms in Europe [11]; thus, the development of effective diagnostics for these organisms is urgently needed.

Diagnosis of TCD in the field, without the support of a specialized equipment, is challenging and not straightforward. Symptoms of GM infection are rather nonspecific and inconspicuous in the early stages of the disease, frequently being confused with abiotic damage, thus hampering diagnosis through visual observation [12]. This is quite common in black walnut stands, as *J. nigra* prefers deep and well-drained neutral soils of fertile alluvial deposits [13]. Since this species has been extensively planted in a wide variety of sites of contrasting moisture, temperature, and edaphic conditions, including humid valley floors, dry ridges, and slopes, it frequently suffers from water stagnation, water scarcity, heat stress, drought, or unsuitable soil [14]. These environmental stress factors represent important constraints to black walnut cultivation, inducing in stands growing at unsuitable sites a generalized symptomatology in many ways identical to that induced by TCD [15].

Morphology-based identification of GM can also be troublesome. The unavailability of a specific substrate or of antibiotics suitable to suppress the growth of contaminating Gram-positive and Gram-negative bacteria and of mycobacteria [16] hampers the recovery of the fungus from environmental samples. A plethora of fungi are also associated with symptomatic woody tissue (underbark necrotic areas around beetle holes and inside galleries) of walnut in the field, and these contaminants often develop in culture faster than the etiologic agent, overgrowing it and masking its presence. GM is in fact slow-growing, taking 3–4 weeks to develop an appreciable colony on an agarized medium, on which it also exhibits considerable phenotypic variation. Furthermore, this ascomycete does not form observable sexual reproductive structures (fruiting bodies) in the outer host tissue or inside beetle galleries; its identification relies upon microscopic examination of the sole vegetative mycelium and conidiophores bearing metulae, phialides, and conidia [17]. Mycological laboratory practice and experience must be, therefore, coupled with time and work to grow, isolate in purity, observe in culture, and obtain morphological confirmation of GM [12,18].

Difficulties inherent with traditional field/conventional (mycological) laboratory diagnoses can be overcome by molecular techniques. Nucleic acid-based diagnostic methods have today become invaluable tools for the early detection of even minute amounts of plant pathogens at an early stage of infection and from a variety of matrices [19]. With particular regard to TCD, there have been recent developments in molecular diagnosis of both the fungus and its insect vector. Oren et al. [18] utilized species-specific microsatellite loci to set up a molecular protocol to detect both GM and PJ from host tissues. The authors of [20] developed target-specific TaqMan-based real-time PCR assays for the detection of 10 harmful pathogens from environmental samples including GM. In these studies, the presence of the fungus was investigated on drilled bark samples, or on artificially inoculated greenhouse grown plants and on the insect vector. The authors of [12], using simplex/duplex qPCR assays, were able to detect GM and its vector PJ from a variety of matrices including pure fungal mycelium, infected tissue, adults of the vector, and their frass. Lastly, the authors of [21] improved pre-existing diagnostic assays used to detect both the fungus and its vector from field samples, developing a user-friendly and cost-effective protocol based on traditional gel electrophoresis (conventional gel and blue-light visualization with modifications) and TaqMan probes. In the present work, we report two diagnostic protocols based on loop-mediated isothermal amplification (LAMP) [22,23] and real-time quantitative PCR (qPCR), based on SYBR Green chemistry, to accurately detect the TCD microbial causal agent from infected plant tissue. Compared to existing protocols, our methods provide new assay formats for the rapid accurate diagnosis of *G. morbida* from environmental samples, with the aim of offering the LAMP technique as a specific, portable detection tool to assist in field TCD surveillance.

## 2. Results

### 2.1. Nucleic Acid Extractions from Mycelium and Woody Samples

Nucleic acid extraction protocols used in this study allowed us to obtain average values of DNA concentration for fungal mycelium and for woody samples of 68.5 ± 2.6 ng·µL^−1^ (A260/280 = 1.98 ± 0.2) and 48.62 ± 4.5 ng·µL^−1^ (A260/280 = 2.01 ± 0.1), respectively. Amplification of the COX plant gene with the LAMP protocol [24] confirmed the successful DNA extraction from woody samples, showing an amplification time (Tamp) of 14.2 ± 2.4 min/s (mean ± standard deviation—SD).

### 2.2. Diagnostic Specificity and Accuracy

All the assays carried out on target and nontarget DNA samples gave positive amplification for all GM isolates from different geographic origins, while no non-GM species gave positive amplification (Table 1). For each GM sample analyzed, amplification curves and single melting peaks were displayed regardless of the starting matrix (fungal mycelium/woody tissue), confirming the specificity of both LAMP and SYBR Green qPCR assays. Mean values of amplification time for rtLAMP were 8.57 min and 14.21 min for GM mycelium and GM-infected woody tissue, respectively (Figure 1a). The SYBR Green qPCR assay showed a Cqs mean of 23.56 for fungal mycelium and 27.94 for woody GM-infected tissue (Figure 2a). The Tm was obtained both from woody tissue and GM mycelium as 90.50 ± 0.5 °C for rtLAMP (Figure 1b) and 87.5 ± 0.5 °C for SYBR green qPCR (Figure 2b). Specificity of vLAMP (visual LAMP) and rtLAMP (real-time LAMP) showed comparable results (Table 1). For each molecular protocol, diagnostic specificity and relative accuracy were found to be 100%. The SYBR Green qPCR protocol results were comparable to those obtained with LAMP (Table 1 and Figure 1c), showing 100% diagnostic specificity.

### 2.3. Blind Panel Validation

The blind panel test with the three techniques used in this study showed a full correspondence between expected and observed values. Therefore, the specificity, sensitivity, and accuracy of the results were 100%. In the case of the rtLAMP assay, the mean T_amp_ (min/s) values recorded on the GM samples were equal to 9.45 ± 0.22 SD from fungal mycelium and 13.56 ± 0.35 SD from infected woody tissue. Visual analysis with the naked eye confirmed the results obtained with the rtLAMP. The accuracy of the previous assays was also confirmed by SYBR Green qPCR assay which gave mean values of Cq equal to 23.79 ± 0.06 and 26.54 ± 0.10, respectively, for fungal mycelium and for infected woody tissue.

### 2.4. Repeatability and Reproducibility of Diagnostic Assays

Repeatability and reproducibility values were based on the standard deviation (SD), and they ranged between 0.01 and 0.52 for the rtLAMP assay and between 0.01 and 0.40 for the SYBR Green qPCR (Table 2).

### 2.5. LAMP and qPCR Sensitivity Assays

The limit of detection (LoD) was 3.2 pg/µL in the rtLAMP, vLAMP, and SYBR Green qPCR assays. The mean T_amp_ (min/s) of rtLAMP at the dilution of 3.2 pg/µL was 16.49 ± 0.57 SD, while, at the same dilution, the Cq of SYBR Green qPCR resulted in a mean value of 35.91 ± 0.49 SD. This LoD value is the same as that of a previously described qPCR probe assay targeting the same fungus [12] (Table 3 and Figure 3a,b).

## 3. Discussion

The molecular assays developed here represent the first LAMP-based assays for the detection of GM. They provide valuable and robust support for the rapid, accurate, and repeatable detection of GM, and they can overcome some limitations of traditional diagnosis. Molecular detection methods allow an accurate diagnosis of GM in a short time even in the absence of specific symptoms. Reliable LAMP results were obtained on woody tissues in a few minutes (ca. 10 min for highly concentrated target DNA), an obvious advantage for speeding up the decision-making process and application of control measures. The molecular assay presented here, is highly specific and also allows detection of the DNA of the target ascomycete in DNA mixtures (DNA from the host tree and/or the plant microbiome). This excludes the possibility that the presence of other microorganisms, frequently present in the infected tissue, may prevent the diagnosis. The molecular methods developed in this work can significantly reduce the time of diagnosis compared to morphological identification, which appears technically demanding, requires experienced staff [25], and is time-consuming, especially when large numbers of samples need to be processed.

This approach complements our previous study [26] on a LAMP-based molecular method to detect PJ. The accurate identification of GM is a pressing need because this fungus is not always associated with its main vector, as observed for many other *Geosmithia* species [27,28]. GM in fact has been also found on 18 other insect species, including ambrosia beetles, bark beetles, and other weevils [29]. The presence of GM on other insects, especially on bark and ambrosia beetles, may complicate TCD epidemiology. In fact, especially in areas where black walnut grows in stressed conditions, secondary native insects may exploit stressed walnuts and disperse the fungus. These circumstances are frequent in Europe, where extensive *J. nigra* plantations have been established at heterogenous sites, many of which are unsuitable for black walnut, as a result of EU funding [30].

Compared to other molecular assays recently developed for the diagnosis of GM and its vector PJ [18,20,21,29], our investigation provides a more accurate and versatile detection, relying on three different methodological approaches. Our molecular toolkit can be exploited both in routine screening (vLAMP, rtLAMP) and in diagnostic confirmation efforts (qPCR Sybr Green, rtLAMP). As previously observed, technical and operational parameters can also be assessed from a qualitative point of view to compare the efficacy of each developed method [31]. vLAMP is the fastest and cheapest method, the easiest to perform, and the most user-friendly; however, it suffers from potentially unclear positive identifications and a greater risk of contamination (false positives). The rtLAMP is more expensive than vLAMP but has the advantage of discriminate specific/aspecific detections by melting peaks and considerable analytical sensitivity. Nevertheless, rtLAMP requires greater technical expertise, as well as dedicated instrumentation. The more expensive technique, SYBR Green qPCR, combines the advantages of intercalation (melting point) with considerable sensitivity. However, it requires dedicated instrumentation and technical skills, and it is more time-consuming than LAMP-based detection. In conclusion, LAMP assays are highly sensitive and promising for on-site diagnosis. They can be used with both endpoint (vLAMP) and real-time LAMP; moreover, they are very fast, taking less than 2 h from DNA extraction to results [32]. The validation of each LAMP assay on the portable instrument performed in this work allowed each assay to be considered field-deployable when coupled with a rapid, user-friendly, and efficient DNA extraction method.

Critical issues for newly developed diagnostic tests, to ensure the accuracy of detection, are the optimization of quality and quantity of DNA extractions, as well as validation parameters (specificity and sensitivity of diagnostic assays). The protocol used in this study, previously applied on various organic matrices (insect individuals, insect frass, wood chips, etc.) [12], is fast (DNA extraction took about 50 min to process up to 24 single samples) and provides good amounts of amplifiable DNA from both fungal mycelium and woody samples. Furthermore, the protocol is cheap and relatively simple to perform in laboratories with only basic equipment. The assessment of the repeatability and reproducibility of the tests revealed excellent performance. In fact, the assays in SYBR Green and rtLAMP had a very low variability (less than 5% in all measurements) [33]. The analytical sensitivity (LoD) was in agreement with previous studies [12].

The molecular tools developed here, based on different techniques targeting different genomic regions, can be used singly or in combination and this guarantees high accuracy of diagnosis, for both the detection of new introductions and the confirmation of the results of the vLAMP detection. This multi-region diagnostic approach, through cross-analysis of identification protocols, reduces false positives, as the different biomolecular techniques can finely complement and validate each other. The good analytical performances of our diagnostic procedures indicate that they can constitute a useful toolkit to detect the presence of the fungus on plant material and other commodities imported from the North American continent and Italy, the only country outside the USA where the disease is currently present.

## 4. Materials and Methods

### 4.1. Collection and Identification of Geosmithia morbida

Following the first identification of a TCD outbreak in the province of Florence [10], a series of inspections were carried out, starting from April 2018, in a walnut plantation (mainly of *J. nigra*) aimed at studying GM and its vector PJ. A total of 24 branches (ca. 2–6 cm in diameter and 10–15 cm in length) were collected in September 2018 from six symptomatic plants according to [12]. Each sample was checked for the presence of PJ holes in the bark, as well as tunnels under the bark, by peeling off the branches. Small tissue pieces were sampled close to necroses and insect galleries from each collected branch (Figure 4). Part of this material was used for isolations, and the rest was employed for DNA extraction. Fungal isolation and morphological identification were carried out according to [16]. For molecular identification, DNA was extracted as described in Section 4.3 from pure fungal mycelium obtained from isolations, and the ITS1-5.8S-ITS2 rDNA region was amplified by PCR, using the universal primers for fungi ITS4 and ITS6 [34]. Following purification and sequencing of the amplicons, a BLAST search on GenBank revealed 99% identity of our isolate (GenBank accession number: MH620784) to North American and north Italian GM isolates.

### 4.2. Fungal Strains and Plant Samples Used to Develop the Diagnostic Assays

Both axenic fungal cultures (Table 1) and symptomatic woody tissue samples were used to set up and validate each protocol. The 36 fungal samples used for this work included five strains of the target species GM, two strains of nontarget *Geosmithia* species, and other fungal taxa. All the samples used in this study were selected on the basis of the following criteria: (a) phylogenetic relatedness to the target pathogen; (b) association with host trees in the genus *Juglans*; (c) cosmopolitan distribution and polyphagy, with frequent occurrence on woody hosts; (d) frequent occurrence on walnut trees suffering for TCD, being opportunistic wound- and stress-related dieback and canker pathogens (Table 1). Fungal isolates/DNA samples were obtained from the mycological collections of the University of Florence, the National Research Council (CNR), or the biomolecular collection of the laboratory of the Phytosanitary Service of the Tuscany Region and other Italian universities and research centers.

### 4.3. DNA Extraction

DNA was extracted from each sample described above in duplicate. Briefly, 2% CTAB buffer with subsequent purification was used with the Maxwell^®^ RSC PureFood GMO purification kit and Authentication Kit provided with the automated purificator MaxWell 16 (Promega, Madison, WI, USA) according to the manufacturer’s instructions. Modifications were introduced in the lysis step of the extraction protocol in relation to the type of matrix as follows: (i) for symptomatic woody tissue, ca. 1 g of matrix was homogenized by means of 10 mL steel jars using a TissueLyzer (Qiagen) for 20 s at 3000 oscillations per minute; (ii) for fungal mycelia, ca. 100 mg were collected and ground in 1.5 mL Eppendorf microtubes using micropestles. An initial purification step, using 2% CTAB buffer (2% CTAB, 2% PVP-40, 100 mM Tris—HCl, pH 8.0, 1.4 M NaCl, 20 mM EDTA, and 1% sodium metabisulfite), was carried out by adding 7 mL to woody tissue and 1 mL to mycelia. The subsequent extraction steps followed [12]. Extracted DNA assessment (quantification and contamination degree) was performed using the QiaExpert (Qiagen) instrument. DNA was eluted in 100 µL of nuclease-free water and stored at −20 °C until use. An rtLAMP reaction targeting the cytochrome oxidase (COX) plant gene was carried out to check the performance of the DNA extracted from plant tissue, following conditions described by [24].

### 4.4. Design of the LAMP and SYBR Green qPCR Primers

Two sets of primers, one for LAMP-based assays and another for a SYBR Green qPCR assay, were designed to specifically target GM. LAMP primers were designed using LAMP Designer software (OptiGene Limited, Horsham, UK) on the basis of consensus sequences of the kinesin gene (accession number KF947525; NCBI database).

The SYBR Green qPCR primers were designed on the beta-tubulin gene (accession number KF853905; NCBI database) using OligoArchitect^TM^ Primers and Probe Online software (Sigma-Aldrich, St. Louis, MO, USA), with the following specifications: a 100 to 380 bp product size; a T_m_ (melting temperature) of 55 to 65 °C; primer length from 16 to 28 bp; absence of secondary structure, when possible. All primers were provided by Eurofins Genomics (Ebersberg, Germany). Primer sequences are shown in Table 4. The specificity of the designed LAMP and SYBR Green qPCR primers (Table 4) was further tested using BLAST^®^ (Basic Local Alignment Search Tool; http://www.ncbi.nlm.nih.gov/BLAST, accessed on 13 July 2021) [35] software. Moreover, to assess the in silico specificity of our primers, sequences with similarity to the GM LAMP and SYBR Green qPCR amplicons were retrieved from the GenBank database and aligned with the MAFFT software implemented in Geneious 10.2.6 [36], set with default parameters (Figure 5, Figure 6 and Figure 7). Unrooted phylogenetic trees from GenBank sequences of GM isolates and related species for the LAMP and SYBR Green qPCR protocols were constructed (Figure 6 and Figure 8, respectively) using Geneious 10.2.4 according to the neighbor-joining method and the Tamura–Nei model with 1000 bootstrap replicates.

### 4.5. LAMP and SYBR Green qPCR Assay Conditions

The rtLAMP reactions were performed on a CFX96 thermocycler (Biorad, Berkeley, CA, USA). Each rtLAMP isothermal amplification was performed according to Abdulmawjood et al. [37] with modifications: at 65 °C for 30 min, followed by an annealing analysis from 65 to 95 °C with ramping of 0.5 °C/s. Except where otherwise indicated, each DNA sample was processed in duplicate in a final volume of 20 μL, including in each run a positive control, consisting of samples with target DNA, and a negative control consisting of pure water (instead of DNA). The rtLAMP reaction mixture was as follows: 10 μL of Isothermal Master Mix OptiGene (ISO-001), 2 μL of primer mixture (0.2 μM of F3/B3, 0.4 μM of LoopF/LoopB and 0.8 μM of FIP/BIP), and 2 μL of template DNA (5 ng·µL^−1^). Positivity of rtLAMP assays was assessed by evaluating amplification minutes (Tamp) and melting temperatures (Tm). vLAMP reactions were performed with the same primers designed for the rtLAMP using the Bst 3.0 DNA polymerase kit (New England Biolabs). vLAMP isothermal amplification was carried out at 65 °C for 30 min, followed by an additional cycle of 80 °C for 2 min, in a 20 μL final volume. The vLAMP mixture was as follows: 2 μL of Isothermal Buffer 10×, 0.6 mM dNTPs, 2 mM MgSO_4_, 0.15 mM HNB, 0.2 M betaine, final concentrations of the LAMP primers 0.2 μM for F3/B3, 0.4 μM for LoopF/LoopB, 0.8 μM for FIP/BIP, 0.32 U/μL Bst 3.0, and 2 μL of template DNA (5 ng·µL^−1^). The hydroxynapthol blue (HNB) dye was included in the reaction mixture to visualize vLAMP results [38]. The positivity of vLAMP reactions was checked at the end of the isothermal amplification by assessing with the naked eye the color change (from violet to blue) of each tested sample (Figure 1). SYBR Green qPCR reactions were performed and optimized by different operators in three thermal cyclers: CFX96 (Biorad, Hercules, CA, USA), Aria MMX (Agilent, Santa Clara, CA, USA), and Rotor Gene (Qiagen, Hilden, Germany). DNA samples were assayed in three replicates, including in each run no-template controls with pure water that served as the negative control and DNA template used as the positive control. SYBR Green qPCR reactions were performed in a final volume of 20 μL. Each tube contained 0.4 µM of both forward and revers primer, 10 μL of Quanti Nova Mastermix SYBR Green (Qiagen, Hilden, Germany), and 2 μL of DNA template. The PCR protocol was as follows: 95 °C (2 min), 40 cycles at 95 °C (10 s), followed by a melting analysis from 65 to 95 °C with ramping of 0.5 °C/s.

### 4.6. Specificity and Accuracy of the LAMP and SYBR Green qPCR Assays

All the fungal strains DNA (Table 1) were tested with each technique to verify the diagnostic specificity of developed assays. Positive results of rtLAMP reactions were assessed by analyzing the time of amplification (T_amp_) and the melting temperature (T_m_) of each tested sample. These parameters were determined according to the diagnostic EPPO standard PM7/98-4 [39].

### 4.7. Blind Panel Validation of the Assays

An internal blind panel test was performed on DNA extracted from 12 samples of GM symptomatic woody samples and 12 DNA samples from different fungal strains (two from *Geosmithia* sp., three from *G. obscura*, three from *G. langdonii*, and four from *Botryosphaeria dothidea*). The test was carried out using the SYBR Green qPCR, rtLAMP, and vLAMP protocols. All DNA samples were previously diluted to a final concentration of 5 ng/µL. Samples were tested in triplicate, including no-template controls (NTCs) (nuclease-free water) as negatives. On the basis of the blind panel results, true positives, false positives, true negatives, and false negatives were assessed according to the validation parameters required by the EPPO standard PM7/98-4 [39].

### 4.8. Repeatability and Reproducibility

Repeatability and reproducibility tests were carried out on GM DNA (sample F125) retrieved from the same woody tissue samples above by applying eight independent DNA extractions. The intra-run variation (repeatability) and the inter-run variation (reproducibility) were assessed through standard parameters (Tamp and its standard deviation (SD) for rtLAMP; Cq and its standard deviation (SD) for SYBR Green qPCR). Each DNA sample was processed in triplicate, diluted to a final concentration of 5 ng·µL^−1^, and tested in two separate series. The mean value and standard deviation were calculated for each sample and for each series of samples, to estimate the repeatability. The reproducibility and the repeatability of each protocol were assessed by comparing the data of two series of samples processed by two different operators on different days [32,40].

### 4.9. Sensitivity

The detection limits of rtLAMP, vLAMP, and SYBR Green qPCR protocols were obtained using 1:5 serial dilutions of GM DNA extracted from woody tissue (from 10 ng/µL to 2.38 fg/µL). To simulate the *in planta* detection limit, an additional test was performed. First, GM DNA (5 ng·µL^−1^) was mixed with plant DNA (from healthy *J. nigra* tissue, at a final concentration of 50 ng·µL^−1^). Serial dilutions (1:5) of DNA from fungal mycelium (from 10 ng·µL^−1^ to 5.12 fg·µL^−1^) in DNA extracts from *J. nigra* woody tissue were made and amplified in triplicate.

## 5. Conclusions

The LAMP (real-time and visual) and SYBR Green qPCR protocols developed in this study can be applied to TCD with rapid and reliable diagnostic results. These assays could be valuable in assisting diagnostic end-users in the (a) control of the import/export of material (plants for planting, wood with bark, wood packaging material, etc.), (b) detection of GM over different territories (walnut plantations, spontaneous individuals, ornamental plant nurseries, etc.) to promptly and effectively eradicate the initial disease foci, and (c) wide-ranging monitoring and surveillance campaigns in territories where TCD is locally present. This work adds to other already established molecular protocols, with the ultimate goal of developing, validating, and promoting common and harmonized diagnostic procedures to be shared by the laboratories in charge of official tests.

## Figures and Tables

**Figure 1 plants-11-01239-f001:**
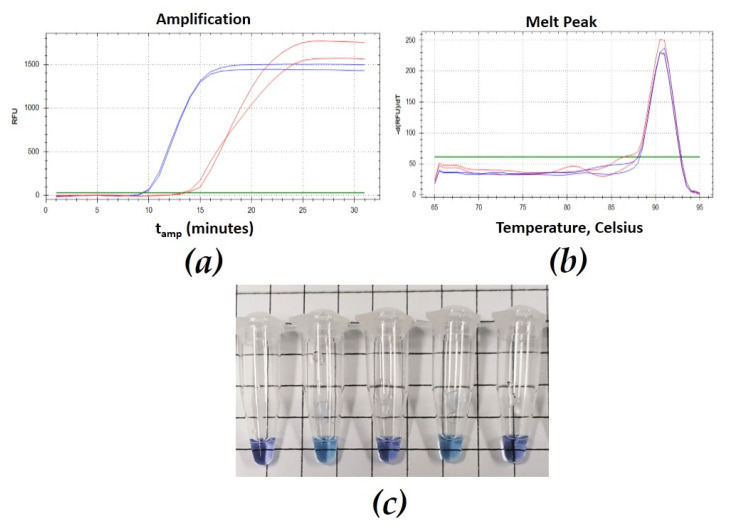
Selection of kinetics of rtLAMP amplification curves (**a**) and melting peaks (**b**) obtained by testing *G. morbida* mycelium (blue) and infected woody tissue (red); (**c**) an example of vLAMP results: dark blue/violet can be assessed as positive, while light blue can be assessed as negative.

**Figure 2 plants-11-01239-f002:**
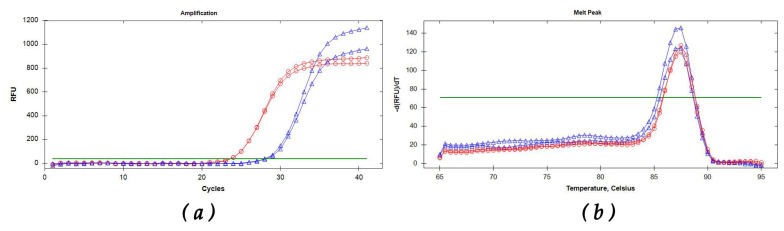
Selection of kinetics of SYBR Green qPCR amplification curves (**a**) and melting peaks (**b**) from *G. morbida* fungal mycelium (red and circles) and infected woody tissue (blue and triangles).

**Figure 3 plants-11-01239-f003:**
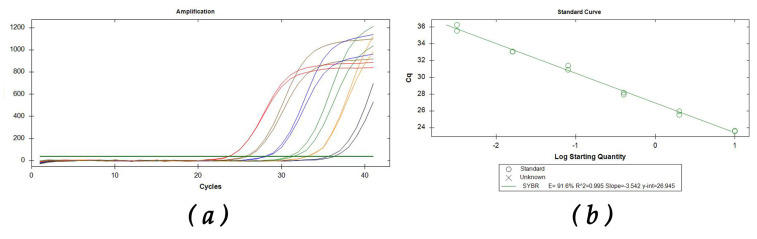
SYBR Green qPCR. Serial dilution 1:5 from 10 ng of DNA extracts from woody tissue infected with *G. morbida*. (**a**) Amplification curves; (**b**) standard curve. The correlation values *r*^2^ were equal to 0.99, and the slope of the standard curve was equal to 3.54.

**Figure 4 plants-11-01239-f004:**
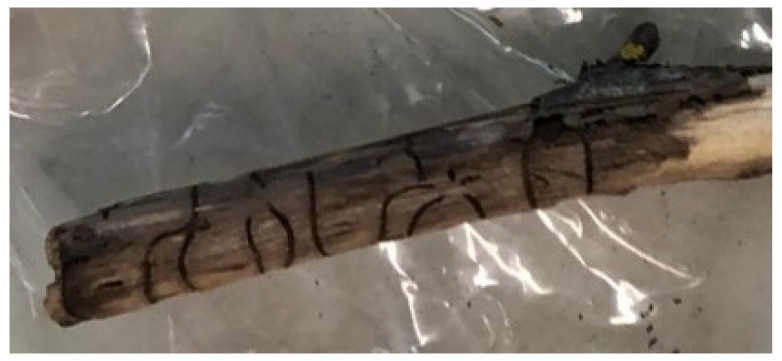
One of the symptomatic small branches of *J. nigra* bearing evident *P. juglandis* galleries utilized for sampling of *G. morbida*-infected tissue.

**Figure 5 plants-11-01239-f005:**
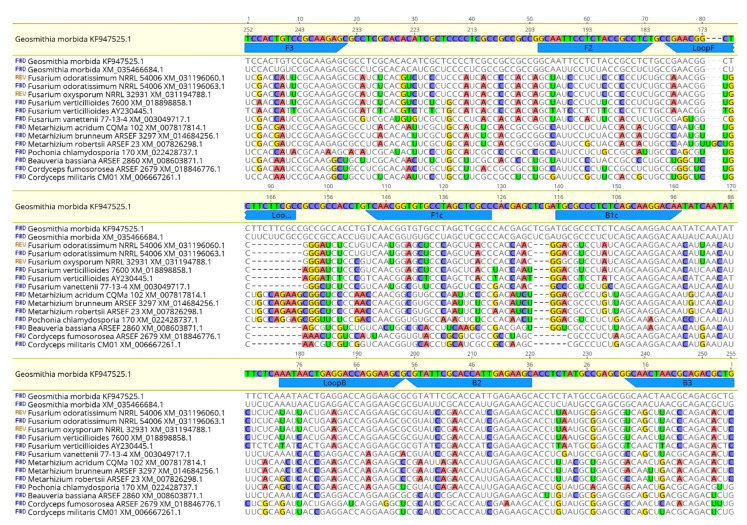
Alignment of a portion of the kinesin gene resulting from the in silico LAMP amplicon of *G. morbida* and similar sequences in GenBank. The alignment is arranged on three levels to better visualize the differences between the *G. morbida* sequence and homologous sequences in GenBank. LAMP primer binding sites are indicated at the top of the figure.

**Figure 6 plants-11-01239-f006:**
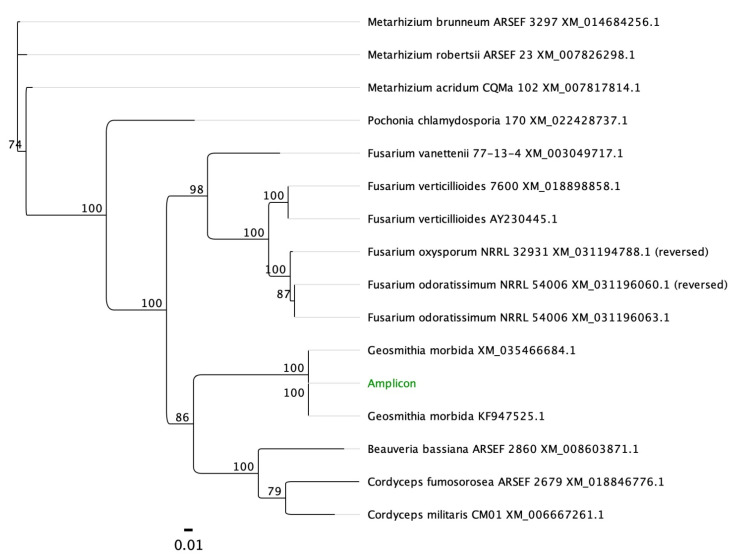
Unrooted phylogenetic tree from Genbank sequences from a portion of the kinesin gene of *G. morbida* isolates and related species for the LAMP protocol. The phylogenetic tree was constructed using Geneious 10.2.4 according to the neighbor-joining method and the Tamura–Nei model with 1000 bootstrap replicates. The Genbank accession numbers of each sample are indicated after the taxon name.

**Figure 7 plants-11-01239-f007:**
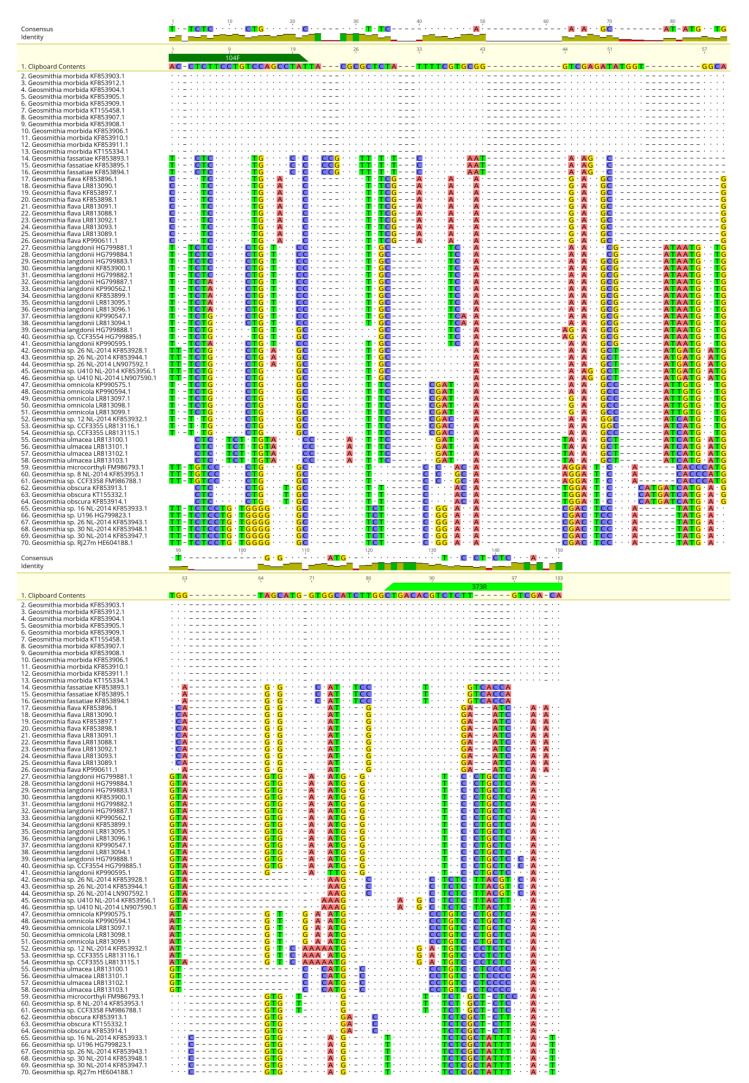
Alignment of a portion of the beta-tubulin gene resulting from the in silico qPCR amplicon of *G. morbida* and homologous sequences in GenBank. The alignment is visualized on three levels to better visualize the differences between the *G. morbida* sequence and homologous sequences in GenBank. The qPCR primer binding sites are shown at the top of the figure.

**Figure 8 plants-11-01239-f008:**
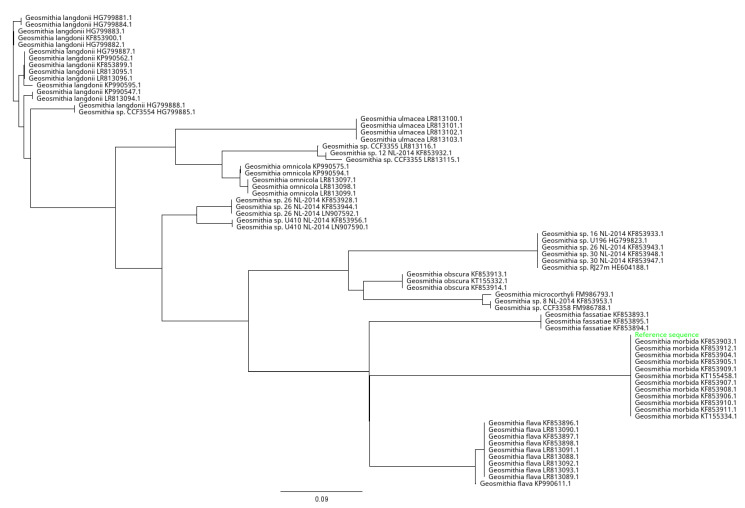
Unrooted phylogenetic tree based on the beta-tubulin gene GenBank sequences of *G. morbida* isolates and related congeneric species for SYBR Green qPCR protocol. The phylogenetic tree was constructed using Geneious 10.2.4 according to the neighbor-joining method and the Tamura–Nei model with 1000 bootstrap replicates. The sequence GenBank accession numbers are reported.

**Table 1 plants-11-01239-t001:** List of target and nontarget fungi from different host species and the results of LAMP and qPCR amplification.

Species	Sample Code	Origin	DNA Extraction Matrix	Supplier	Host	LAMP Results		qPCR
						vLAMP	rtLAMP	SYBR Green
*Geosmithia morbida* (M. Kolařík, Freeland, C. Utley & Tisserat)	F120	Italy	Mycelium	University of Florence	*Juglans nigra*	*+*	*+*	*+*
	F116	Italy	Mycelium	SFR phytopathol. Lab	*Juglans nigra*	*+*	*+*	*+*
	F133	Colorado, USA	Mycelium	CBS—124664	*Juglans nigra*	*+*	*+*	*+*
	F120	Italy	Infected woody tissue	SFR phytopathol. Lab	*Juglans nigra*	*+*	*+*	*+*
	F121	Italy	Infected woody tissue	SFR phytopathol. Lab	*Juglans nigra*	*+*	*+*	*+*
	F124	Italy	Infected woody tissue	SFR phytopathol. Lab	*Juglans nigra*	*+*	*+*	*+*
	F125	Italy	Infected woody tissue	SFR phytopathol. Lab	*Juglans nigra*	*+*	*+*	*+*
	F126	Italy	Infected woody tissue	SFR phytopathol. Lab	*Juglans nigra*	*+*	*+*	*+*
	F127	Italy	Infected woody tissue	SFR phytopathol. Lab	*Juglans nigra*	*+*	*+*	*+*
	F128	Italy	Infected woody tissue	SFR phytopathol. Lab	*Juglans nigra*	*+*	*+*	*+*
	F129	Italy	Infected woody tissue	SFR phytopathol. Lab	*Juglans nigra*	*+*	*+*	*+*
*Geosmithia obscura* (M. Kolařík, Kubátová & Pažoutová)	CCF3422	Czech Republic	Mycelium	ASCR-Prague	−	*−*	*−*	*−*
*Geosmithia langdonii* (M. Kolařík, Kubátová & Pažoutová)	CNR105	Italy	Mycelium	IPSP-CNR-Florence	−	*−*	*−*	*−*
*Geosmithia* sp. (Pitt)	CNR132	Italy	Mycelium	IPSP-CNR-Florence	−	*−*	*−*	*−*
*Botryosphaeria* (Ces. & De Not.) sp.	F004	Italy	Mycelium	SFR phytopathol. Lab	−	*−*	*−*	*−*
*Botryosphaeria dothidea* (Moug.) Ces. & De Not.	F005	Italy	Mycelium	University of Bologna	−	*−*	*−*	*−*
*Sphaeropsis sapinea* ((Fr.) Dyko & B. Sutton)	F078	Italy	Infected plant tissue		−	*−*	*−*	*−*
*Colletotrichum* (Corda)	F010	Italy	Mycelium	SFR phytopathol. Lab	−	*−*	*−*	*−*
*Colletotrichum acutatum* (J.H. Simmonds)	F013	Italy	Mycelium	SFR phytopathol. Lab	−	*−*	*−*	*−*
*Colletotrichum coccodes* (Wallr.) S. Hughes	F011	Italy	Mycelium	University of Bologna	−	*−*	*−*	*−*
*Colletotrichum gloeosporioides* (Penz.) Penz. & Sacc.	F012	Italy	Mycelium	SFR phytopathol. Lab	−	*−*	*−*	*−*
	F102	Italy	Infected plant tissue	SFR phytopathol. Lab	*Photinia* sp.	*−*	*−*	*−*
*Fusarium oxysporum* (Schltdl.)	F026	Italy	Mycelium	University of Bologna	−	*−*	*−*	*−*
*Fusarium oxysporum* f. sp. *radicis-lycopersici* (Jarvis & Shoemaker)	F001	Italy	Mycelium	SFR phytopathol. Lab	*Solanum lycopersicum*	*−*	*−*	*−*
*Fusarium oxysporum* f. sp. *dianthi* (W.C. Snyder & H.N. Hansen)	F123	Italy	Mycelium	University of Pisa		*−*	*−*	*−*
*Fusarium oxysporum* f. sp. *basilici* (Tamietti & Matta)	F122	Italy	Mycelium	University of Pisa		*−*	*−*	*−*
*Fusarium redolens* (Wollenw.)	F027	Italy	Mycelium	University of Bologna	−	*−*	*−*	*−*
*Gibberella circinata* (Nirenberg & O’Donnell)	F028	Italy	Mycelium	CREA-PAV_Rome	−	*−*	*−*	*−*
*Neofusicoccum luteum* (Pennycook & Samuels) Crous, Slippers & A.J.L. Phillips	F038	Italy	Mycelium	University of Florence	*Vitis* sp.	*−*	*−*	*−*
*Neofusicoccum parvum* (Pennycook & Samuels) Crous, Slippers & A.J.L. Phillips	F039	Italy	Mycelium	University of Florence	*Vitis* sp.	*−*	*−*	*−*
*Neofusicoccum ribis* (Slippers, Crous & M.J. Wingf.) Crous, Slippers & A.J.L. Phillips	F040	Italy	Mycelium	University of Florence	*Vitis* sp.	*−*	*−*	*−*
*Neofusicoccum vitifusiforme* (Van Niekerk & Crous) Crous, Slippers & A.J.L. Phillips	F041	Italy	Mycelium	University of Florence	*Vitis* sp.	*−*	*−*	*−*
*Guignardia citricarpa* (Kiely)	F029	Italy	Infected plant tissue	SFR phytopathol. Lab		*−*	*−*	*−*
	F109	Italy	Infected plant tissue	SFR phytopathol. Lab		*−*	*−*	*−*
	F114	Italy	Mycelium	SFR phytopathol. Lab -Campania		*−*	*−*	*−*
	F115	Italy	Infected plant tissue	SFR phytopathol. Lab -Campania		*−*	*−*	*−*
*Guignardia mangiferae* (A.J.Roy)	F030	Italy	Mycelium	University of Bologna		*−*	*−*	*−*
*Dothiorella sarmentorum* ((Fr.) A.J.L. Phillips, A. Alves & J. Luque)	F147	Italy	Mycelium	University of Florence		*−*	*−*	*−*
*Phyllosticta paracitricarpa* (Guarnaccia & Crous)	F134	Greece	Mycelium	CBS 141358		*−*	*−*	*−*
*Drechslera teres* ((Sacc.) Shoemaker)	F022	Italy	Mycelium	University of Bologna		*−*	*−*	*−*
*Biscogniauxia mediterranea* ((De Not.) Kuntze)	F100	Italy	Mycelium	University of Florence		*−*	*−*	*−*
*Phomopsis* sp. (Sacc. & Roum.)	F046	Italy	Mycelium	University of Bologna		*−*	*−*	*−*

**Table 2 plants-11-01239-t002:** Repeatability and reproducibility of real-time LAMP and SYBR Green qPCR assays on infected woody tissue, measured as standard deviation (SD).

Sample No.	rtLAMP			SYBR Green qPCR		
	Repeatability		Reproducibility	Repeatability		Reproducibility
	Assay 1	Assay 2		Assay 1	Assay 2	
F120	0.02	0.03	0.25	0.40	0.28	0.30
F121	0.01	0.25	0.35	0.06	0.35	0.23
F124	0.02	0.10	0.24	0.02	0.10	0.20
F125	0.18	0.13	0.18	0.29	0.18	0.20
F126	0.52	0.09	0.46	0.11	0.13	0.12
F127	0.52	0.45	0.36	0.01	0.11	0.09
F128	0.37	0.15	0.24	0.19	0.03	0.13
F129	0.06	0.06	0.29	0.28	0.01	0.19

**Table 3 plants-11-01239-t003:** LoD assay based on *G. morbida* DNA from woody tissue using 1:5 serial dilutions (from 10 ng·µL^−1^ to 5.12 fg·µL^−1^). The data are also compared with the same samples used for the qPCR Probe assay [12]. In the vLAMP column, positive (+) and negative (−) results are indicated.

Dilutions	rtLAMP	vLAMP	SYBR Green qPCR	qPCR Probe *G. morbida* [12]
	Tamp (Mean ± SD)		Cq (Mean ± SD)	Cq (Mean *±* SD)
10 ng·µL^−1^	8.90 ± 0.15	+	23.60 ± 0.06	25.54 ± 0.48
2.0 ng·µL^−1^	9.58 ± 0.63	+	25.75 ± 0.30	27.52 ± 0.43
0.4 ng·µL^−1^	10.74 ± 0.25	+	28.06 ± 0.17	29.57 ± 0.47
0.08 ng·µL^−1^	11.41 ± 0.98	+	31.16 ± 0.35	31.12 ± 0.14
0.016 ng·µL^−1^	13.02 ± 1.03	+	33.08 ± 0.01	33.05 ± 0.53
3.2 pg·µL^−1^	16.49 ± 0.57	+	35.91 ± 0.49	34.75 ± 0.19
0.64 pg·µL^−1^	−	−	−	−
0.128 pg·µL^−1^	−	−	−	−
0.0256 pg·µL^−1^	−	−	−	−
5.12 fg·µL^−1^	−	−	−	−

**Table 4 plants-11-01239-t004:** LAMP and SYBR Green qPCR primers designed for *Geosmithia morbida*.

Molecular Assay	Target Gene	Primer Name	Sequence 5′–3′	Lenght (bp)	Amplicon Size (bp)
LAMP	kinesin gene	F3	TCCACTGTCCGCAAGAGC	18	166
		B3	CAGCGTCTGCGTTAGTTGC	19	
		FIP(F1c+F2)	GGCGAGCTAGGCACACCGTTGAGCAATTCCTCTACCGCCTCT	42	
		BIP(B1c+B2)	TGCGCCCTCTCAGCAAGGACAAGCTTCTCAATGGTGCGAATACG	44	
		LoopF	GCGAAGAAGAGCCGTTCG	18	
		LoopB	AATAACTGAGGACCAGGAAGCG	22	
SYBR Green qPCR	beta-tubulin gene	Gmorb_104_F	ACCTCTTCCTGTCCAGCCTAT	21	269
		Gmorb_373_R	TGTCGACAAGAGACGTGTCAG	21	

## Data Availability

The data presented in this study are available on request from the corresponding author.

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
