# Peer review of "Loop-Mediated Isothermal Amplification (LAMP) and SYBR Green qPCR for Fast and Reliable Detection of Geosmithia morbida (Kolařik) in Infected Walnut"

_plants, 2022, doi:10.3390/plants11091239_

Round 1
Reviewer 1 Report
This is a carefully done study and the findings are of considerable interest. A few minor revisions are listed below.
Some photos of vLAMP should be included in this paper.
Line 28: “Thousand Canker Disease” should read “Thousand Cankers Disease”.
Line 147: Provide a note on a asterisk in Table 1 need description.
Line 147: Several “+” and “-“ are out of alignment.
Line 143: The full name of “vLAMP” should be clearly stated at least once.
Line 143: The full name of “rtLAMP” should be clearly stated at least once.
Line 180: “G.morbida” should be Italic.
Line 180: The column for vLAMP in Table 3 are out of alignment.
Line 180: “-“ in the column of SYBR Green aPCR are out of alignment.
Line 180: Table 3 lacks the result of qPCR probe G. morbida [12].
Line 191: “A)” and “b)” should read “(a)” and “(b)”, respectively.
Line 229: “aspecific” is “specific”?
Line 366: The LAMP assay [37] was run at 65ºC for 30 min with a melting curve analysis step (annealing curve 98°–80°C ramping at 0.1 per min) in a portable real-time fluorometer (Genie II, Optigene, UK). The annealing analysis of this paper is different with that of Abdulmawjood et al. (2014).
Line 411: Specify the GM sample code.

Author Response
Response (in bold) to:
REVIEWER 1
Comments and Suggestions for Authors
This is a carefully done study and the findings are of considerable interest. A few minor revisions are listed below.
Some photos of vLAMP should be included in this paper.
Done
Line 28: “Thousand Canker Disease” should read “Thousand Cankers Disease”.
Done
Line 147: Provide a note on a asterisk in Table 1 need description.
Done
Line 147: Several “+” and “-“ are out of alignment.
Done
Line 143: The full name of “vLAMP” should be clearly stated at least once.
Done
Line 143: The full name of “rtLAMP” should be clearly stated at least once.
Done
Line 180: “G.morbida” should be Italic.
Done
Line 180: The column for vLAMP in Table 3 are out of alignment.
Done
Line 180: “-“ in the column of SYBR Green aPCR are out of alignment.
Done
Line 180: Table 3 lacks the result of qPCR probe G. morbida [12].
Line 191: “A)” and “b)” should read “(a)” and “(b)”, respectively.
Done
Line 229: “aspecific” is “specific”?
This sentence has been rewritten (more clearly)
Line 366: The LAMP assay [37] was run at 65ºC for 30 min with a melting curve analysis step (annealing curve 98°–80°C ramping at 0.1 per min) in a portable real-time fluorometer (Genie II, Optigene, UK). The annealing analysis of this paper is different with that of Abdulmawjood et al. (2014).
This sentence was amended writing that the protocol of Abdulmawjood et al. (2014) with modifications has been followed
Line 411: Specify the GM sample code.
Done

Reviewer 2 Report
The paper of Rizzo et al. reports the development of LAMP and SYBR Green assays for the detection Geosmithia morbida (Kolařik) in infected walnut. The results obtained from the experiments of this research are interesting and merit publication. However, I would suggest the following correction\clarifications, prior to publication:
- I would suggest modifying the title to read: Loop‑mediated isothermal amplification (LAMP) and SYBR Green qPCR for fast and reliable detection of Geosmithia morbida (KolaÅ™ik) in infected walnut (there is no need to specify “tissue” since the experiment has not delt solely with plant tissue but also isolated mycelium.
- L140: Correct to read: SYBR Green
- What I didn’t appreciate very much in this experiment is the use of JUST ONE infected plant tissue sample, infected with GM, for the development of the diagnostic assays. Given that this work is addressed to diagnosticians more than to researchers, in general, and to detect GM from infected plants rather than from the artificial media (mycelium), I would have included instead more plant samples for the standardization of the developed techniques. This was not done! and whether due to the lack of infected plant tissue and\or this issue has skipped the authors’ attention, I cannot tell! However, I would indicate in the text at least the period of sampling carried out in the field to address the reader over an approximate biological period for sampling this pathogen.
- Authors has reported in Table 1 a list of target and non-target fungi\templates, all used for the standardization of their assays, whereas in Figure 1, four GM-positive reactions are showed as amplification curves and ONLY 1 or 2 curves for the negative templates! Authors should have included in the figure at least 10 different non-target samples to corroborate their claim on the specificity of their tests.
- In Table 2; it is not clear what the 8 samples listed stand for? This experiment was constructed ONLY on 4 GM-positive templates! So what are the others samples?
- It’s not clear how authors have stabilized the value of 3.2 microgram as an end of the detection limit?
- Why 2 different LAMP assays (rtLAMP and vLAMP) were developed; so, give some justification.
- All DNA extraction methods used by the authors are somehow laborious, when applied in the field. So, the claim of using these methods and related assays in the field is somehow speculative and not totally correct. This is true if authors have directly utilized the infected plant tissue without any prior DNA extraction, as reported in other similar fungal pathogens. Thus, moderate your descriptions in the text on the applicability and easiness of performing these assays, when are related to a field application. However, in the lab, I don’t see any difference between 1 hour or 30 min of DNA extraction phases that worth any mentioning.
Author Response
Response (in bold) to:
REVIEWER 2
Comments and Suggestions for Authors
The paper of Rizzo et al. reports the development of LAMP and SYBR Green assays for the detection Geosmithia morbida (Kolařik) in infected walnut. The results obtained from the experiments of this research are interesting and merit publication. However, I would suggest the following correction\clarifications, prior to publication:
I would suggest modifying the title to read: Loop‑mediated isothermal amplification (LAMP) and SYBR Green qPCR for fast and reliable detection of Geosmithia morbida (KolaÅ™ik) in infected walnut (there is no need to specify “tissue” since the experiment has not delt solely with plant tissue but also isolated mycelium.
Done
L140: Correct to read: SYBR Green
Done
What I didn’t appreciate very much in this experiment is the use of JUST ONE infected plant tissue sample, infected with GM, for the development of the diagnostic assays. Given that this work is addressed to diagnosticians more than to researchers, in general, and to detect GM from infected plants rather than from the artificial media (mycelium), I would have included instead more plant samples for the standardization of the developed techniques. This was not done! and whether due to the lack of infected plant tissue and\or this issue has skipped the authors’ attention, I cannot tell! However, I would indicate in the text at least the period of sampling carried out in the field to address the reader over an approximate biological period for sampling this pathogen.
We corrected this by specifying that we worked on 6 symptomatic plants (24 samples in total)
Authors has reported in Table 1 a list of target and non-target fungi\templates, all used for the standardization of their assays, whereas in Figure 1, four GM-positive reactions are showed as amplification curves and ONLY 1 or 2 curves for the negative templates! Authors should have included in the figure at least 10 different non-target samples to corroborate their claim on the specificity of their tests.
The specificity results are reported in Table 1. Here we have deemed it appropriate to report only a selection of amplification curves obtained from both mycelium and woody tissues
In? This experiment was constructed ONLY on 4 GM-positive templates! So what are the others samples?
We actually worked on 24 tissue samples taken from 6 different symptomatic plants (this has now also been added in the text). The 8 samples listed in Table 2 (which had been numbered from 1 to 8 in the former version) had been collected during a single sampling in the field. Now they have been coded each with its own code (see also Table 1)
It’s not clear how authors have stabilized the value of 3.2 microgram as an end of the detection limit?
The explanation we have given in the paragraph 4.9 Sensitivity (Materials and Methods) seems sufficiently clear to us.
Why 2 different LAMP assays (rtLAMP and vLAMP) were developed; so, give some justification.
LAMP is simple but when applied in real-time version, it needs instruments that can read the fluorescence emitted by DNA amplification during the reaction. In an on-site application perspective, the more the assay is user-friendly and the equipment is free the more the technique can be deployable to non-specialized users. Visual LAMP allows to determine the amplification results directly in post-amplification with naked eye, not requiring specialized skills neither expensive equipment. Since naked-eye visualization has become a routine approach concerning LAMP diagnostics, we thought it useful to also report this approach as an alternative to rtLAMP. This with the aim to rise awareness among possible end-users of the possibility to use vLAMP as a cheaper method.
All DNA extraction methods used by the authors are somehow laborious, when applied in the field. So, the claim of using these methods and related assays in the field is somehow speculative and not totally correct. This is true if authors have directly utilized the infected plant tissue without any prior DNA extraction, as reported in other similar fungal pathogens. Thus, moderate your descriptions in the text on the applicability and easiness of performing these assays, when are related to a field application.
According to your suggestion, we have moderated our descriptions in the text on the applicability and easiness of performing these assays in the field. However, we had already highlighted these difficulties in the text with the sentence below (end of paragraph):
“The validation of each LAMP assay on the portable instrument performed in this work allowed each assay to be considered field-deployable when coupled with a rapid, user-friendly, and efficient DNA extraction method”.
However, in the lab, I don’t see any difference between 1 hour or 30 min of DNA extraction phases that worth any mentioning.
We have not made any time comparison regarding the extraction times (half an hour vs 1 hour). We have only reported the timing of our extractions highlighting that they are fast.

Reviewer 3 Report
The manuscript is well-written. I recommend the publication of the manuscript after minor revisions.
Please see my comments below:
- Update the referencing style. For example, in page 3, line 106, “ [18] utilized species-specific microsatellite loci to set…..” Replace “[18]” by “last name of the first author et. al [18]”
- In figure 1a, correct the title of the x-axis.
- For figure 3a, use different color lines for different DNA concentrations
Author Response
Response (in bold) to:
REVIEWER 3
Comments and Suggestions for Authors
The manuscript is well-written. I recommend the publication of the manuscript after minor revisions.
- Update the referencing style. For example, in page 3, line 106, “ [18] utilized species-specific microsatellite loci to set…..” Replace “[18]” by “last name of the first author et. al [18]”
Done
- In figure 1a, correct the title of the x-axis.
Done
- For figure 3a, use different color lines for different DNA concentrations
Done
